# Neoadjuvant Strategies for Patients with Resectable Biliary Tract Cancers: A Review

**DOI:** 10.3390/curroncol32100584

**Published:** 2025-10-20

**Authors:** Chelsea R. Olson, Gabriela L. Aitken, Michael W. Spinrad, Evan S. Glazer

**Affiliations:** 1Department of Surgery, College of Medicine, The University of Tennessee Health Science Center, 910 Madison Ave., Suite 325, Memphis, TN 38163, USA; colson18@uthsc.edu (C.R.O.); gaitken1@uthsc.edu (G.L.A.); mspinrad@uthsc.edu (M.W.S.); 2Center for Cancer Research, College of Medicine, The University of Tennessee Health Science Center, Memphis, TN 38163, USA

**Keywords:** cholangiocarcinoma, neoadjuvant therapy

## Abstract

**Simple Summary:**

Cholangiocarcinoma (CC) is a deadly cancer that arises from the epithelial cells of the biliary tree. Biliary tract cancers (BTC) include both CC and gall bladder cancer. Surgical resection is considered the only curative treatment. Recently, treatments for patients based on a molecularly targeted approach have improved survival rates. In patients with local or limited regional disease, neoadjuvant therapies offer a way to shrink tumors and evaluate cancer spread over a short period. There are very few data evaluating neoadjuvant (pre-surgery) strategies in patients with resectable disease. The goal of this comprehensive mini-review is to summarize the data and provide a rationale for the role of neoadjuvant treatment in patients with resectable BTC. Although there is no high-level evidence, studies show that neoadjuvant therapy that incorporates targeted treatments and immunotherapies under multidisciplinary oversight benefits select patients and is a valuable tool in the treatment of BTC.

**Abstract:**

Cholangiocarcinoma (CC) is a rare and aggressive malignancy that arises from the epithelial cells (cholangiocytes) of the biliary tree. Biliary tract cancers (BTC) include both CC and gall bladder cancer. Surgical resection is considered the only curative treatment. Recently, however, a fundamental shift in the understanding of the molecular profiles of these tumors has led to a molecular-targeted approach with improved survival rates in some patients with these tumors. In patients with local or limited regional disease, neoadjuvant therapies offer a way to downstage tumors, assess tumor biology, potentially achieve R0 resection, and potentially prevent both locoregional and distant recurrence by treating occult micrometastatic disease. Because BTC are rare and surgery is the standard of care for patients with non-metastatic disease, there is very little data evaluating neoadjuvant strategies in resectable disease. Immunotherapies and molecularly targeted agents originally developed for advanced disease in the adjuvant or palliative settings are now being considered for neoadjuvant use. This review aims to summarize the data and provide a rationale for the role of neoadjuvant treatment in patients with resectable BTC. While there is no high-level evidence, studies show that neoadjuvant therapy that incorporates targeted treatments and immunotherapies under multidisciplinary oversight benefits select patients and is a valuable tool in the treatment of BTC. We favor molecular testing to guide neoadjuvant therapy for patients with BTC, when feasible, to prevent unnecessary operations and minimize the risk of recurrence or metastasis.

## 1. Introduction

Biliary tract cancers (BTC) comprise four distinct cancer types based on location: intrahepatic cholangiocarcinoma (iCC), perihilar cholangiocarcinoma (pCC, extrahepatic bile ducts at the confluence of the hepatic ducts), distal cholangiocarcinoma (dCC, distal to the confluence of the hepatic ducts to the ampulla), and gallbladder carcinoma (GBC) [1,2]. Despite their embryological relationships, each site of BTC has unique biological, clinical, and molecular differences (Figure 1). These cancers are often grouped together because of their cell of origin (cholangiocytes) and the overall rarity of each cancer. This review aims to better delineate the current potential treatment strategies for patients with resectable (or likely resectable) BTC. This review provides a broad and critical overview of the existing literature.

Cholangiocarcinoma (CC) accounts for approximately 3% of all gastrointestinal cancers, making it a rare malignancy [3]. Each subtype (iCC, pCC, and dCC) has distinct epidemiological, molecular, and surgical considerations. The American Joint Committee on Cancer (AJCC) defines BTC staging. It has been incorporated into standard practice by the National Comprehensive Cancer Network (NCCN) guidelines, with stratification into resectable, borderline resectable, and locally advanced/unresectable categories to guide treatment selection and surgical planning (Table 1) [2]. Locally advanced BTC is characterized by malignant lymphadenopathy on imaging, vascular invasion, or bilobar hepatic disease. This is typically stage IIB or III (Table 1). Resectability does not rely on tumor size alone but also on the location of the tumor in relation to vital structures, multifocality even within the same lobe of the liver, and future liver remnant.

Complete surgical resection is considered the only mechanism to achieve a cure; however, greater than 70% of patients present with locally advanced or unresectable disease. Resectability does not rely on tumor size alone but also on the location of the tumor in relation to vital structures, multifocality even within the same lobe of the liver, and future liver remnant. Consensus multi-disciplinary tumor board evaluation to identify resectability is the bedrock to characterize patients with non-metastatic BTC.

In general, BTC has a poor prognosis, with a 5-year survival rate of 26% for pCC and 36% for extrahepatic disease overall for all patients. Recurrence is common after resection, indicating the potential value of systemic therapy in these patients for treating occult micrometastatic disease [4,5,6]. Based on these observations, many experts have recommended neoadjuvant therapy. Following the ABC-02 trial [7], the standard of care became adjuvant gemcitabine and cisplatin, which demonstrated improved overall survival. Recently, progress in the development of immunotherapy and targeted agents has broadened the options for advanced disease. With the exciting new trend of achieving pathologic complete response (pCR) in other malignancies treated with neoadjuvant therapies, including immunotherapy, many are looking to apply the same treatment techniques to other cancers with high mortality rates. In exceptionally rare cases, immunotherapy for cholangiocarcinoma can induce a pCR to achieve cure without additional treatment [2,4,8,9,10]. Typically, before effective adjuvant therapies were identified, neoadjuvant therapy was expected to be futile; therefore, upfront attempts at resection were the norm [5].

Neoadjuvant systemic therapy offers the potential to initiate treatment sooner, downstage tumors, assess tumor biology preoperatively, and increase R0 resection rates, particularly in borderline resectable cases [10,11,12,13,14]. Since the NCCN guidelines are evidence-based and the evidence is lacking, neoadjuvant therapy is not recommended for all patients with CC. An older retrospective series by Ercolani et al. showed that in their cohort of 72 patients, only 35% received adjuvant chemotherapy due to reasons such as deconditioning or complications after surgery, but this rate has increased more recently [15]. Thus, neoadjuvant regimens allow patients to benefit from systemic treatment prior to potential deconditioning or complications after surgery. Based on the high rate of metastatic and recurrent disease in patients with CC, this strategy allows for the treatment of occult micrometastatic disease. It not only evaluates patient fitness to tolerate major resection but also allows for theoretically improved treatment delivery to the malignancy prior to vascular supply disruption by surgery [5,12,13]. These rationales have led to the recent consideration of neoadjuvant therapies in the NCCN Guidelines for Biliary Tract Cancers. This review aims to summarize the current data on neoadjuvant therapy in patients with BTC, provide a framework for future studies, and offer reasonable treatment strategies for clinicians. The rationale is to provide data and recommendations to clinicians on the best use of neoadjuvant therapy for patients with BTC. The clinical question we seek to address is whether neoadjuvant therapy improves survival in patients with non-metastatic BTC.

## 2. Materials and Methods

We searched PubMed and Scopus for English language articles using the search terms “neoadjuvant therapy,” “cholangiocarcinoma,” and “bile duct cancers.” We evaluated the articles for the type and quality of evidence. The search was limited to the Enlish language and full text availability. Case studies and case reports were excluded. Comprehensive literature reviews are exempt as non-human subject research by the UTHSC Institutional Review Board. The articles were obtained and evaluated by at least two authors with descrepencies decided upon by the corresponding author. The outcomes included disease site, type of study, type of therapy, differences in survival, and rationale for specific treatment strategies.

All studies with complete data were evaluated and listed in the References section. We utilized the GRADE framework [16]. Studies and data within a given article were grouped based on the disease site and type of treatment being evaluated. Biases and limitations were assessed based on the study design, limitations listed in each article, and by the authors of this article.

## 3. Results

### 3.1. Current Data on the Role of Neoadjuvant Therapy

Neoadjuvant therapy often precedes surgical resection by 4–6 months based on treatment duration and response to therapy [4,13,17,18]. Recent neoadjuvant regimens of cytotoxic agents and immunotherapy have demonstrated efficacy in decreasing tumor size and enabling margin-negative resection in many patients, with improved survival [14,19]. The effectiveness of neoadjuvant therapies (chemotherapy, immunotherapy, and radiotherapy), as well as their benefits and risks, will be examined independently for each disease site.

There are limited data on the use of neoadjuvant radiotherapy in patients with resectable CC [20]. We identified 336 potential studies (October 2025). The vast majority were non-contributary (e.g., limited case reports or case series) or did not yield sufficient data for any outcome of interest. Overall, 45 studies were used in this manuscript. We did not identify any high-quality prospective datasets or large randomized trials on the use of neoadjuvant radiotherapy in patients with CC. The evidence is GRADE level low. We excluded ‘very-low-quality’ evidence (GRADE framework). While there are ongoing studies, some data suggest that adjuvant chemoradiotherapy and immunoradiotherapy improve survival; however, these studies are relatively small [21,22]. The receipt of adjuvant radiotherapy seems to be most beneficial for patients with node-positive disease [23]. This creates significant equipoise, but the data supporting systemic therapy for aggressive CC are generally of higher quality than those supporting radiotherapy treatment.

Overall, we recommend some use of neoadjuvant therapy for all patients. Figure 1 delineates the best available strategies for consideration in context of available literature and probability of having actionable results for an individual patient. Systemic chemotherapy has the most data available while radiotherapy may be considered for GBC and pCC given recent results and ongoing trials.

### 3.2. Intrahepatic Cholangiocarcinoma

Currently, the NCCN recommends upfront surgery for non-metastatic, resectable iCC, in part due to the limited high-level evidence. There is no standard neoadjuvant chemotherapy (NAC) for the treatment of patients with iCC, and no phase III studies have evaluated survival in patients with iCC who receive NAC. Retrospective series, reviews, and prospective phase II trials have suggested NAC as a reliable downstaging strategy in iCC [3]. Utuama et al. reported a median overall survival of 47.6 months vs. 25.9 months and 5-year OS rates of 34% vs. 25.7% (log-rank, *p* = 0.10) with NAC compared to immediate surgery in a propensity-matched analysis of high-risk (Stage II and III), upfront-resectable iCC [24].

The GAP regimen was adapted from a phase II study by Shroff et al. [25] which added nab-paclitaxel to gemcitabine and cisplatin in the treatment of advanced cholangiocarcinoma. The single-arm phase II NEO-GAP trial confirmed the feasibility of a neoadjuvant triplet regimen (gemcitabine, cisplatin, and nab-paclitaxel) for resectable, high-risk iCC. 16.7% of patients showed radiologic progression of disease on restaging scans, and 6.7% of patients were found to have metastatic disease at the time of surgery. There were no treatment-related mortalities, and 73% of the patients were able to complete both chemotherapy and surgery. Importantly, the study also showed that surgical complications were not different in their treatment group compared to those who did not receive NAC.

A retrospective national population-based comparative cohort study by Parente et al. [19] found an association between NAC and increased overall survival (OS) for all non-metastatic cholangiocarcinoma subtypes compared to surgery alone. However, it did not show any difference between OS for non-metastatic iCC patients receiving NAC and those receiving adjuvant chemotherapy (AC) (HR 1.19, 95% CI 0.99–1.45, *p* = 0.068), suggesting that the receipt of both resection and chemotherapy was the primary driver of survival advantage. The results of this study suggest that the sequence of treatment is less important than receiving all possible treatments.

Immunotherapy for iCC is usually reserved for unresectable or metastatic disease [2,14,26]. However, there has been an increased focus on the incorporation of immunotherapy, targeted therapies, and their combinations in neoadjuvant regimens for non-metastatic iCC. [27] As iCC is molecularly heterogeneous (common mutations *ARID1A*, *BAP1*, *BRAF*, *ERBB2* (HER2), *FGFR2*, *IDH1/2*, *KRAS*, *MDM2*, *SMAD4*, and *TP53*), there is more potential for targeted treatments [28]. A recent review [28] of molecularly targeted treatments by Gujarathi et al. highlighted that molecular profiling of iCC is integral to its management, especially in unresectable or metastatic settings.

Liver-directed therapies in the neoadjuvant period are more likely to be recommended for patients who are poor surgical candidates or have an extensive disease burden. Thermal ablation is generally reserved for rare patients with small single tumors (T1a) less than 3 cm or 4 cm in size in high-risk locations or prohibitibve comorbiditeis. Arterial directed options like bland embolization, transarterial chemoembolization (TACE), and Y-90 transarterial radioembolization (TARE) may be combined with chemotherapy for possible downstaging or as palliative treatment [14,29]. A hepatic arterial infusion (HAI) pump may be placed in specialized centers or in specific trials for bilobar hepatic disease, but this treatment is not routinely used [30]. There is very little data regarding its use or ability to achieve disease free status in large groups of unselected patients [4,30]. Ongoing trials are evaluating the use of systemic therapy (as standard of care) with HAI to determine if there is a survival benefit.

### 3.3. Perihilar Cholangiocarcinoma

To date, there have been no multicenter prospective studies investigating neoadjuvant therapies prior to resection in the treatment of perihilar or hilar cholangiocarcinoma. Typically, these are grouped together in treatment strategy. However, protocols for neoadjuvant chemoradiation therapy in conjunction with liver transplantation initially designed at the Mayo Clinic and University of Nebraska have been used for decades [31,32]. A multicenter retrospective study found that recurrence-free survival at 2, 5, and 10 years was 78% (95% CI 72–84), 65% (95% CI 57–73), and 59% (95% CI 49–69), respectively, across 12 centers with established protocols for liver transplantation with neoadjuvant chemoradiation for unresectable disease [33]. Similar studies conducted recently have reported similar results [34,35].

The role of neoadjuvant therapy prior to surgical resection has been less studied. Early reports from McMasters et al. [36] and Nelson et al. [37] demonstrated improved R0 resection rates in patients with borderline or unresectable disease who received neoadjuvant chemoradiation, but with no survival benefit. Jung et al. [38] performed a retrospective cohort study comparing neoadjuvant chemoradiation prior to surgery versus upfront surgery. There was a significant increase in downstaging (91.7% vs. 51.1% *p* = 0.01). However, there were no significant differences in R0 resection (83.3% vs. upfront surgery 64.4%, *p* = 0.32), microvascular invasion (50% vs. upfront surgery 64.4%, *p* = 0.51), and lymph node metastasis (25% vs. upfront surgery 55.6%, *p* = 0.06). There was also no improvement in disease-free or overall survival, although it should be noted that the neoadjuvant cohort contained only 12 patients, and the study may be underpowered [38].

### 3.4. Distal Cholangiocarcinoma

NAC and chemoradiation for dCC have been evaluated retrospectively. To date, no prospective randomized controlled trials have been conducted. Cloyd et al. [8] reported no difference in 5-year OS between dCC patients who received preoperative therapy and those who did not (46.6% vs. 49.1%, *p* > 0.05). However, the actual receipt of either neoadjuvant or adjuvant therapy was associated with improved OS [8]. Furthermore, lymph node positivity was the strongest predictor of OS, suggesting that neoadjuvant therapy may play a role in cases with high-risk features, such as lymph node involvement. A more recent study by Parente et al. [19] using the National Cancer Database (NCDB) found that 7.2% of patients undergoing surgery for dCC received neoadjuvant therapy. In adjusted analyses, these patients had significantly improved OS compared with surgery alone (HR 0.65, 95% CI 0.53–0.78, *p* < 0.001). However, sensitivity analyses demonstrated no difference in OS between patients receiving neoadjuvant or adjuvant chemotherapy after surgery for dCC (HR 1.13, 95% CI 0.91–1.41, *p* = 0.264).

### 3.5. Gallbladder Carcinoma

Current NCCN guidelines recommend considering NAC for stage T1b or greater GBC since these cancers present incidentally at the time of cholecystectomy and immediate re-operation carries higher risks, but there was not complete consensus on this recommendation. There is limited clinical trial data to define a standard regimen or definitive benefit; there is no preferred regimen listed in the NCCN guidelines, but it is suggested to be the same as that for unresectable and metastatic disease.

A systematic review by Hakeem et al. [10] (474 total patients) found that approximately one third of patients achieve an R0 resection with the use of NAC or neoadjuvant chemoradiotherapy. Chaudhari et al. [39] (160 patients) found a response rate of 52.5% and a clinical benefit rate of 70% in patients with GBC treated with NAC, with 41.2% of patients undergoing resection with curative intent. A small prospective study by Engineer et al. [40] (n = 28 patients) showed that an R0 resection was achieved in 50% of patients with locally advanced GBC treated with neoadjuvant chemoradiation with good local control (93%) and 5-year survival (47%), suggesting that radiation has potential in the treatment of GBC in the neoadjuvant setting. A phase III randomized controlled trial (POLCAGB) by Engineer et al. [41] is underway to compare NAC with neoadjuvant chemoradiation in patients with locally advanced GBC. Early data (abstract only) suggests that neoadjuvant chemoradiotherapy increases overall survival and event-free survival compared to neoadjuvant chemotherapy.

## 4. Discussion

***Overall strategy, limitations, and gaps in the literature***. Overall, the choice of neoadjuvant treatment is based on the location of the tumor in or around the liver (Figure 1) and personalized to the patient’s performance status utilizing any actionable molecular data. We favor triplet therapy (as found in the NCCN guidelines) due to the aggressive nature of CC, although this approach does run the risk of “overtreatment” of some patients. The greatest gap in the literature regarding current treatment strategies for patients with BTC remains the lack of data which is also the greatest limitation in this work. Other major limitations include study heterogeneity, patient heterogeneity, reporting/publication biases, and limitations to molecular data analysis. The authors biases towards neoadjuvant therapy should also be acknowledged.

We firmly believe that all patients should be presented at a multidisciplinary conference to obtain input from medical oncologists, surgical oncologists, radiation oncologists, and other team members. Specific treatments, both in the neoadjuvant setting and surgery, are often limited by comorbidities and patient physiology. A more nuanced understanding of the probability that a given patient will actually be able to undergo resection prior to neoadjuvant therapy will help determine whether the treatment is truly neoadjuvant or definitive (and potentially palliative in this context). Unfortunately, tumor biology in each of these locations is unique which discounts the ability to use molecular targeted therapy or immunotherapy in all settings [13,14,42,43,44,45]. Although data support the use of biomarker-driven treatment in many patients with locally advanced BTC, data on its use in patients with upfront resectable disease are limited [29,42,43]. While retrospective comparisons are problematic to apply to other patient populations, they offer mechanisms to identify patient populations that may benefit from improved patient selection. We hypothesize that neoadjuvant therapy is a tool to better select patients with resectable BTC who will benefit from this treatment strategy but this is also an intrinsic bias.

Another gap in the literature is that biomarker-driven treatments and studies require a certain amount of tissue to be obtained. Often a too limited amount of tissue is obtained for molecular profiling for individuals limiting the ability to identify new biomarkers or studies or actionable biomarkers in patients. Liquid biopsy (circulating tumor DNA [ctDNA], and/or ctRNA) techniques can be used to molecularly characterize BTC and identify potential treatment options [42,43]. While the cost efficacy and benefits on survival with these assays remains unknown, they offer a unique opportunity to offer targeted therapy in the neoadjuvant setting based on a rational approach, personalized therapy, and much larger specimens obtained at the time of resection for novel studies. Our approach is to use ctDNA and tumor molecular profiling in the neoadjuvant setting for patients who have higher than average surgical risk and those with larger tumors that approach locally advanced disease on multi-disciplinary review. Low-quality studies remain on ongoing with limited published manuscripts; however, growing data suggest that ctDNA offers very helpful data when actionable results are identified. This, of course, is obvious *a priori*, but still an important area of further research.

***Molecularly Targeted Therapies***. For patients with metastatic disease, there is significant evidence that patients benefit from targeted therapy based on molecular tumor testing (profiling). This was previously referred to as next generation sequencing but now includes many more tests than genetic sequencing. For CC, these include molecular aberrations such as NTRK gene fusion, deficient mismatch repair proteins, BRAF mutation, FGFR2 fusion/rearrangements, IDH1 mutations, HER2 overexpression/amplification, RET gene fusion, and *KRAS* mutation [15].

In the neoadjuvant setting, molecular testing is of benefit for patients with high-risk ICC given the likelihood of metastatic disease being found at surgery or during the neoadjuvant course [28]. For patients with high-risk dCC, such as lymphadenopathy, we extrapolate the potential benefit for systemic therapy given that cancer has already spread regionally. Our practice is to perform molecular panel testing given the significant overlap of molecular aberrations across the various sites of BTC. There is less data regarding molecular testing for neoadjuvant therapy for patients with pCC, but given the very high-risk nature of the disease, we also recommend panel molecular testing when able to do so.

***Intrahepatic Cholangiocarcinoma***. There are retrospective cohort analyses, reviews, and prospective phase II trials showing the feasibility of neoadjuvant treatment for nonmetastatic iCC. Although there are no trials showing efficacy, there are many theoretical benefits to neoadjuvant therapies for resectable and borderline resectable iCC including opportunity to receive complete course of systemic therapy before potential postoperative complications or deconditioning that may prevent a patient from starting or completing systemic treatments. Considering some studies show as little as 35% of patients receive adjuvant therapy due to postoperative complications or deconditioning [15], better strategies are needed to ensure systemic therapy is given to most of these paitents. The cohort study by Parente et al. suggests that neoadjuvant administration of systemic treatment would provide survival benefit to patients with nonmetastatic iCC by raising the probability that systemic therapy is actually received by the patient [19].

Therefore, multidisciplinary discussions about each patient’s disease should take place with a strong consideration as to which sequence of therapies are most likely to be successfully delivered to the patient with the least degree of toxicities or induced comorbidities. For those patients deemed to have unresectable (locally-advanced) disease, additional immunotherapeutic strategies are reasonable to attempt to convert the disease status to resectable when molecularly profiling allows for such treatments [2,6,26,43,44].

There is a lack of high evidence data on the use of NAC for iCC patients, similar to other rare cancers. There is a risk that the neoadjuvant regimen chosen may be ineffective or inferior to another regimen which would allow for progression of disease during neoadjuvant treatment. This might result in a potential missed opportunity to achieve surgical R0 resection. There are possible side effects of neoadjuvant regimens that may interfere with the patient’s ability to endure a major liver resection. Additionally, there are possible effects of neoadjuvant therapy on ease of resection, and postoperative complications such as wound healing. In the case of iCC, NAC may be considered for high-risk diseases, preferably in clinical trial settings.

***Perihilar Cholangiocarcinoma***. Patients who meet criteria for liver transplantation are often offered neoadjuvant chemoradiation prior to transplant, as established by many transplant center protocols. This includes patients with unresectable disease with tumors ≤ 3 cm and no nodal involvement or metastasis. The role of neoadjuvant therapy prior to curative resection remains unknown and further research is needed to recommend a regime. Further, the decision to biopsy a pCC should be made in context of the local transplant center’s approach as transperitoneal biopsy may be a contraindication to transplant.

In evaluation of the evidence for neoadjuvant treatment for pCC, it should be noted that there are limitations. There is significant heterogeneity in what defines hilar cholangiocarcinoma as “unresectable” in the literature and patient populations in these studies are center specific, limiting generalizability. Currently, while neoadjuvant therapy is associated with increased rates of downstaging and possible increase in R0 resection rates, there has not been a survival benefit in patients undergoing liver resection. However, in patients who meet criteria for liver transplantation, there is a role for neoadjuvant chemoradiation, with an increase in overall and disease-free survival compared to patients that do not receive a liver transplant with neoadjuvant therapy. Ethun et al. compared neoadjuvant therapy with transplant vs. curative resection in tumors that qualify for transplant and found a significant increase in OS (5-year survival 64% vs. 18%) associated with transplant in this patient population [46].

Our strategy for pCC is to refer patients to transplant centers to be evaluated for transplant research protocols. For patient ineligible for transplant, we then recommend upfront systemic chemotherapy to evaluate for intra-hepatic metastases given the risk of this finding in the postoperative time period. For patients who do not have a cancer diagnosis despite an aggressive evaluation with endoscopic procedures, we will consider a diagnostic laparoscopy with liver ultrasound and biopsy. This also affords the opportunity to evaluate for resectability with a relatively low risk procedure.

***Distal Cholangiocarcinoma***. Systemic therapy has a clear role in the treatment of dCC, as it improves OS. However, retrospective analyses demonstrate that there is no difference in OS between administration in the neoadjuvant or adjuvant settings suggesting a personalized strategy is ideal. Our bias is to offer systemic therapy in the neoadjuvant setting to assure receipt of treatment. Lymph node positivity is an independent predictor of poor prognosis, which suggests that there may be a role for neoadjuvant therapy in locally advanced, high-risk disease as well. Randomized controlled trials are needed to investigate the optimal timing of therapy; however, overall, we are comfortable with 4–6 cycles of systemic therapy.

Based on the findings above, NAC can be viewed as an alternative to adjuvant therapy. That said, these studies are retrospective and have several limitations. For example, patients selected for surgery after NAC as well as those who remained eligible to receive adjuvant chemotherapy after surgery were likely better overall performers at baseline and, therefore, had improved OS. Additionally, NAC was used in locally advanced disease compared to adjuvant chemotherapy in resectable disease, which shows selection bias and may have impacted the outcomes. Finally, using NAC arguably selects for better tumor biology than not using a NAC strategy.

It is important to note that data for dCC are often mixed with those for pCC and gallbladder carcinoma, which makes it difficult to ascertain whether neoadjuvant therapy truly confers a benefit for patients with dCC. The studies listed here were specific to dCC, allowing for conclusions specific to this disease site. The findings confirm that there is a role for *receipt* systemic therapy in dCC with no survival difference in terms of timing. Therefore, for patients with dCC, we highly recommend a reasonable course of NAC for nearly all patients (3–6 months in duration of doublet or triplet therapy).

***Gallbladder Carcinoma***. The NCCN guidelines provide limited support in the use of neoadjuvant therapy for patients with GBC, albeit without complete consensus. Given the poor prognosis, neoadjuvant systemic therapy should be considered under the guidance of a multidisciplinary team for patients with lymph node positivity or other high-risk features, as it is vital to rule out rapid disease progression and avoid futile surgery. The OPT-IN study (EA2197), which is closed to accrual, was to evaluate the role of NAC for patients with gall bladder cancer but it is unknown if it has accrued enough patients/events to provide definitive answers. Until better data is available, we recommend 4–6 cycles of systemic therapy with a typically aggressive regime (triplet therapy).

Neoadjuvant treatment allows for evaluation of tumor biology, especially when lymphadenopathy is identified, by ruling out rapid progression. This strategy also helps to identify the patients who are most likely to benefit from surgical intervention and avoid futile surgery. The decision to pursue NAC should be individualized for each patient in close consultation with a multidisciplinary team, starting with a period of 2 to 6 months and reassessing every 2 to 3 months pending results of the POLCAGB trial.

## 5. Conclusions

Overall strategies for neoadjuvant therapy in patients with BTC center on optimizing the receipt of effective/active therapy, the risk of occult metastatic disease, and overall patient health. The data was too heterogenous and of too poor quality to perform a meaningful meta-analysis for each type of BTC. No large, randomized trials were found to inform the use of neoadjuvant therapy in this patient population. BTC are spatially distinct as four distinct cancers and molecularly even more diverse, offering an abundance of potential treatments at the population level. While personalized, multidisciplinary care remains paramount, our approach focuses on maximizing systemic therapy, highly selecting patients for surgery who are most likely to benefit, and limiting radiotherapy for specific indications where it is most likely to be beneficial.

## Figures and Tables

**Figure 1 curroncol-32-00584-f001:**
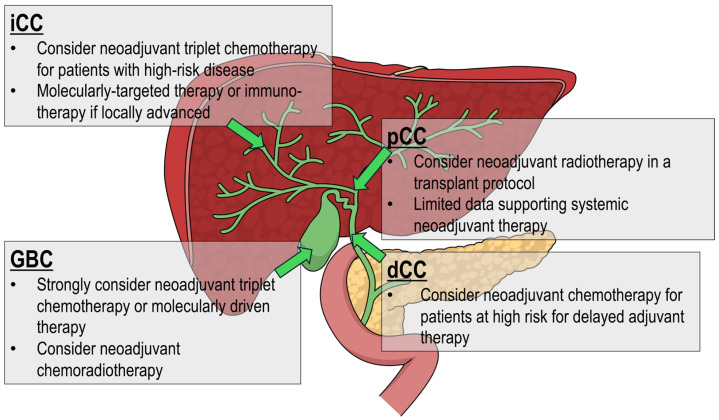
Overall strategy for each type of cholangiocarcinoma. Overall, the specific type of neoadjuvant treatment is based on the tumor location in or around the liver, whereas the specific regimen is personalized to the patient’s performance status. XRT: radiotherapy, iCC: intrahepatic cholangiocarcinoma, pCC: peri-hilar/hilar cholangiocarcinoma. GBC: gallbladder cancer. dCC: extrahepatic cholangiocarcinoma excluding ampullary carcinoma and pCC.

**Table 1 curroncol-32-00584-t001:** Staging of biliary tract cancer by subtype. Based on American Joint Committee on Cancer (AJCC) 8th edition 2017.

Stage	Intrahepatic	Perihilar	Extrahepatic	Gallbladder
0	0 Tis: In situ (intraductal)	0 Tis: In situ	0 Tis: In situ/High Grade Dysplasia	0 Tis: In situ
I	IA T1aN0M0: solitary tumor ≤ 5 cm without vascular invasion	I T1N0M0: invasion into muscle layer but confined to bile ducts.	I T1N0M0: Invades bile duct wall with depth of invasion < 5 mm	I T1N0M0: Invades lamina propria or muscle layer of gallbladder
	IB T1bN0M0: solitary tumor > 5 cm without vascular invasion			
II	II T2N0M0: solitary tumor of any size with intrahepatic vascular invasion or multiple tumors with or without vascular invasion	II T2aN0M0: tumor invades beyond wall of the duct to surrounding adipose tissue T2bN0M0: tumor invades beyond wall of the duct to surrounding hepatic parenchyma	IIA T1N1M0: tumor invades bile duct wall < 5 mm with 1–3 regional lymph nodes positive T2N0M0: tumor invades bile duct 5–12 mm with no nodes positive	IIA T2aN0M0: tumor invades perimuscular connective tissue on peritoneal side without involvement of serosa (visceral peritoneum)
			IIB T2N1M0: tumor invades 5–12 mm with 1–3 regional lymph nodes positive T3N0-1M0: tumor invades > 12 mm with 0–3 regional lymph nodes positive	IIB T2bN0M0: tumor invades the perimuscular connective tissue on hepatic side with no extension into the liver
III	IIIA T3N0M0: tumor perforates visceral peritoneum	IIIA T3N0M0: tumor invades unilateral branches of portal vein or hepatic artery	IIIA T1-3N2M0: any tumor invasion of the bile duct with ≥4 regional lymph nodes positive	IIIA T3N0M0: tumor perforates serosa and/or directly invades liver, and/or one other adjacent organ or structure (stomach, duodenum, colon, pancreas, omentum, extrahepatic bile ducts)
	IIIB T4N0M0: tumor involving local extrahepatic structures by direct invasion T1-4N1M0: Any T stage with any regional lymph metastases present.	IIIB T4N0M0: tumor invades main portal vein or its branches bilaterally, or the CHA, or unilateral 2nd order biliary radicals bilaterally with contralateral portal vein/hepatic artery involvement.	IIIB T4N0-2M0: tumor involves the celiac axis, SMA, and or CHA with any number of nodes (including 0) positive	IIIB T1-3 N1M0: T stage 1–3 with 1–3 regional lymph nodes positive
		IIIC T1-4N1M0: Any T stage with 1–3 regional lymph nodes positive.		
IV	IV T1-4N0-1M1: distant metastasis present	IVA T1-4N2M0: Any T stage with 4 or more positive regional lymph nodes	IV T1-4N0-2M1: distant metastasis present	IVA T4N0-1M0: tumor invades main portal vein, or hepatic artery, or invades 2 or more extrahepatic organs/structures with 0–3 regional lymph nodes positive
		IVB T1-4N0-2M1: distant metastasis present		IVB T1-4N2M0: Any T stage with 4 or more regional lymph nodes positive. T1-4N0-2M1: distant metastasis present

## Data Availability

The dataset supporting the conclusions of this article is publicly available in PubMed.

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
