# Peer review of "Neoadjuvant Strategies for Patients with Resectable Biliary Tract Cancers: A Review"

_curroncol, 2025, doi:10.3390/curroncol32100584_

Round 1
Reviewer 1 Report
Comments and Suggestions for Authors
This study was aimed to review the current status and future perspective of neoadjuvant strategies in patients with resectable biliary tract cancers.
Authors especially addressed the importance of personalized approach due to too heterogenous diseases and also small number of patients of biliary tract malignancies.
1, It might be uncertain whether Fig 1 was required to understand this review. I thought this fig might be not required in this manuscript.
2, As this study reviewed about neoadjuvant issues for only resectable disease of biliary tract cancers, reviewed publications were not so large numbers of publications. Therefore, I thought that the title of this study might be more suitable for “Mini-Review” rather than “Comprehensive Review”.
Author Response
Thank you for reviewing our manuscript. We appreciate the comments and have made the recommended changes. We have also edited the manuscript at length. Please see below.
Comment 1: It might be uncertain whether Fig 1 was required to understand this review. I thought this fig might be not required in this manuscript.
Response 1: Thank you for this comment. It brings up an excellent point- the difference between experts in this space and those who are not. The reviewer is an expert so we have modified the use of the Figure to help non-experts understand anatomic differences in the location of cholangiocarcinomas.
Comment 2: As this study reviewed about neoadjuvant issues for only resectable disease of biliary tract cancers, reviewed publications were not so large numbers of publications. Therefore, I thought that the title of this study might be more suitable for “Mini-Review” rather than “Comprehensive Review”.
Response 2: Agreed. Some of the word choice/description was based on the journal's preference. Overall, we do not feel strongly and do agree with the reviewer. We have changed the title.
Reviewer 2 Report
Comments and Suggestions for Authors
The research is on the management of biliary tract tumours (BTC), a rare but clinically important cancer subtype characterised by a dismal prognosis, with a particular emphasis on neoadjuvant therapy.
The publication goes into great length on the several forms of cholangiocarcinoma, such as intrahepatic, perihilar, and distal cholangiocarcinoma, as well as gallbladder cancer. The publication also has parts that are just about each subtype.
The authors emphasise the need of integrating novel strategies into clinical practice, including molecularly targeted therapies, immunotherapy, and circulating tumour DNA (ctDNA) analysis. This shows how the clinical environment is changing.
The text is organised in a methodical manner, and it has distinct sections for the introduction, methods, results, discussion, and conclusion.
The statement recognises the lack of high-level evidence and stresses the need of making judgements in a thoughtful fashion that considers several fields.
Improvement needed:
Clarity in the Methodology Approach
The method used for reviewing the literature lacks sufficient information. We have found the databases PubMed and Scopus, but we still need to find the PRISMA-style flow, search terms, inclusion and exclusion criteria, and chronology. This makes it difficult to reproduce.
The authors claim that "at least two authors reviewed each article," but they don't say how conflicts were resolved.
The integration of data and the hierarchical framework of evidence.
The review is based largely on retrospective series and a small number of phase II studies. This is OK since these kinds of investigations don't happen very often. However, the article may examine the amount of evidence more closely, perhaps by use the Oxford or GRADE systems.
Even if there wasn't a formal meta-analysis, a table showing the results would make the information clearer. This summary would provide details on the study design, the number of people who took part, the results, and how strong the evidence is.
Evaluation of the Criticality Level
Although the study is descriptive, it might benefit from a more thorough examination of the constraints inherent in prior studies, including the variability in definitions of staging, selection bias, and small sample numbers.
Looking at current clinical research, such POLCAGB for the gallbladder and ctDNA-based investigations, would give us a better idea of what could happen in the future.
Consistency
Some subsections, such intrahepatic cholangiocarcinoma, are quite well-written and include information on current studies. But other subsections, like those on perihilar and distal cholangiocarcinoma, are just as short and don't go into as much detail.
It would be beneficial to establish more precise distinctions, since the discourse often conflates obviously resectable conditions with borderline resectable conditions.
Tables and Figures
There is just one figure, which is a diagram of the overall plan. Adding summary tables that are not required might make a review easier for readers to understand. These tables put data next to each other based on things like where the cancer is, what sort of treatment it is, and how much of a survival benefit it gives.
Conclusions
The conclusion emphasises the scarcity of high-level evidence; however, it could be enhanced by offering more specific and pragmatic recommendations for clinicians to implement (for instance, "In the case of iCC, NAC may be considered for high-risk diseases, preferably in clinical trial settings").
To make the primary point clearer, it would be helpful to include more specific instructions concerning the time and patient selection, as well as the order of treatments in connection to direct resection.
Language
Even if most of the words make sense, there are still a lot of them that are repeated. The phrase "NAC permits systemic therapy prior to surgical deconditioning" appears in many distinct forms.
You may make anything easier to read by taking out unneeded words and making little changes to the style.
The work is well-organised, thorough, and on topic, but I think it needs a lot of changes before it can be published.
Make the process longer and explain the search method in more detail.
Combine structured evidence tables.
Expand the examination of the study's heterogeneity and limitations.
Make practical, therapeutically useful ideas to back up the results.
Get the depth of BTC classes to be equal.
Comments on the Quality of English LanguageEven if most of the words make sense, there are still a lot of them that are repeated. The phrase "NAC permits systemic therapy prior to surgical deconditioning" appears in many distinct forms.
You may make anything easier to read by taking out unneeded words and making little changes to the style.
Author Response
Comment 1: The research is on the management of biliary tract tumours (BTC), a rare but clinically important cancer subtype characterised by a dismal prognosis, with a particular emphasis on neoadjuvant therapy. The publication goes into great length on the several forms of cholangiocarcinoma, such as intrahepatic, perihilar, and distal cholangiocarcinoma, as well as gallbladder cancer. The publication also has parts that are just about each subtype. The authors emphasise the need of integrating novel strategies into clinical practice, including molecularly targeted therapies, immunotherapy, and circulating tumour DNA (ctDNA) analysis. This shows how the clinical environment is changing. The text is organised in a methodical manner, and it has distinct sections for the introduction, methods, results, discussion, and conclusion. The statement recognises the lack of high-level evidence and stresses the need of making judgements in a thoughtful fashion that considers several fields.
Thank you for these comments. We appreciate the feedback.
Comment 2: Improvement needed: Clarity in the Methodology Approach. The method used for reviewing the literature lacks sufficient information. We have found the databases PubMed and Scopus, but we still need to find the PRISMA-style flow, search terms, inclusion and exclusion criteria, and chronology. This makes it difficult to reproduce.
We agree that the Methods section lacked clarity. Therefore, we addressed by better defining terms, clarifying statements, and addressing the reviewers’ critiques.
The authors claim that "at least two authors reviewed each article," but they don't say how conflicts were resolved.
The approach was clarified.
The integration of data and the hierarchical framework of evidence.
This was clarified in the Methods section.
The review is based largely on retrospective series and a small number of phase II studies. This is OK since these kinds of investigations don't happen very often. However, the article may examine the amount of evidence more closely, perhaps by use the Oxford or GRADE systems.
Thank you for this comment and identifying our lack of clarity. The methods and approach were clarified. We also directly addressed aspects of the GRADE framework throughout the manuscript.
Even if there wasn't a formal meta-analysis, a table showing the results would make the information clearer. This summary would provide details on the study design, the number of people who took part, the results, and how strong the evidence is.
Thank you for identifying our lack of clarity. We more precisely summarized the study design, methods, and search results in the manuscript.
Evaluation of the Criticality Level. Although the study is descriptive, it might benefit from a more thorough examination of the constraints inherent in prior studies, including the variability in definitions of staging, selection bias, and small sample numbers.
Thank you for this point. We examined the included studies in a more thorough manner, adding to the Methods and Results sections. We also addressed selection and publication bias more thoroughly.
Looking at current clinical research, such POLCAGB for the gallbladder and ctDNA-based investigations, would give us a better idea of what could happen in the future.
Thank you for this point. We have added this data.
Consistency. Some subsections, such intrahepatic cholangiocarcinoma, are quite well-written and include information on current studies. But other subsections, like those on perihilar and distal cholangiocarcinoma, are just as short and don't go into as much detail.
This reviewer brings up an important point- different sections have different qualities. We agree and fully acknowledge this, but respectfully, point out that much of this is due to difference in the quality of data for the various sites of disease.
With that said, we have enhanced these sections of the manuscript with additional context, strategy, considerations, and recommendations.
It would be beneficial to establish more precise distinctions, since the discourse often conflates obviously resectable conditions with borderline resectable conditions.
We agree with this statement. Overall, we view this a spectrum, often with institutional biases and preferences as to what actually differentiates an unresectable from resectable tumor. We clarified these distinctions more precisely in the manuscript.
Comment 3: Tables and Figures. There is just one figure, which is a diagram of the overall plan. Adding summary tables that are not required might make a review easier for readers to understand. These tables put data next to each other based on things like where the cancer is, what sort of treatment it is, and how much of a survival benefit it gives.
We clarified/enhanced the tables and figures.
Comment 4: Conclusions. The conclusion emphasises the scarcity of high-level evidence; however, it could be enhanced by offering more specific and pragmatic recommendations for clinicians to implement (for instance, "In the case of iCC, NAC may be considered for high-risk diseases, preferably in clinical trial settings").
Thank you for this important comment. We have made the recommend changes.
To make the primary point clearer, it would be helpful to include more specific instructions concerning the time and patient selection, as well as the order of treatments in connection to direct resection.
Thank you for this important comment; we agree. We have made the recommend changes in the Discussion section.
Comment 5: Language. Even if most of the words make sense, there are still a lot of them that are repeated. The phrase "NAC permits systemic therapy prior to surgical deconditioning" appears in many distinct forms.
The manuscript was aggressively edited with changes tracked/marked.
You may make anything easier to read by taking out unneeded words and making little changes to the style.
The manuscript was aggressively edited with changes tracked/marked.
The work is well-organised, thorough, and on topic, but I think it needs a lot of changes before it can be published. Make the process longer and explain the search method in more detail.
Thank you for this feedback. We agree. The manuscript was aggressively edited.
Combine structured evidence tables.
The tables and figures were improved, clarified, and enhanced.
Expand the examination of the study's heterogeneity and limitations.
Thank you for this important comment. We have made the recommend changes.
Make practical, therapeutically useful ideas to back up the results.
This is a good point. We have expanded therapeutic recommendations in the Discussion section for each type of BTC.
Get the depth of BTC classes to be equal.
Thank you for this comment. We agree but believe this is more about the heterogeneity in published data for each disease site. We have expanded out assessment, recommendations, and limitations in each site in order to provide more depth and context to all BTC.
Comment 6: Comments on the Quality of English Language. Even if most of the words make sense, there are still a lot of them that are repeated. The phrase "NAC permits systemic therapy prior to surgical deconditioning" appears in many distinct forms. You may make anything easier to read by taking out unneeded words and making little changes to the style.
Round 2
Reviewer 2 Report
Comments and Suggestions for Authors
Thank you for the modification and explanation. I agree on the final version.